# Consolidation Systemic Therapy in Locally Advanced, Inoperable Nonsmall Cell Lung Cancer—How to Identify Patients Which Can Benefit from It?

**Branislav Jeremić** [1,*], **Elene Mariamidze** [2] , **Inga Shoshiashvili** [3] **and Ivane Kiladze** [3]

1   School of Medicine, University of Kragujevac, 34000 Kragujevac, Serbia
2   Department of Oncology and Hematology, Todua Clinic, 0112 Tbilisi, Georgia
3   Department of Medical Oncology, Caucasus Medical Center, 0186 Tbilisi, Georgia
*   Correspondence: nebareje@gmail.com; Tel.: +381-69-4146663

**Abstract:** Background: Consolidation systemic therapy (ST) given after concurrent radiotherapy (RT) and ST (RT-ST) is frequently practiced in locally advanced inoperable nonsmall cell lung cancer (NSCLC). Little is known, however, about the fate of patients achieving different responses after concurrent phases of the treatment. Methods: we searched the English-language literature to identify full-length articles on phase II and Phase III clinical studies employing consolidation ST after initial concurrent RT-ST. We sought information about response evaluation after the concurrent phase and the outcome of these patient subgroups, the patterns of failure per response achieved after the concurrent phase as well as the outcome of these subgroups after the consolidation phase. Results: Eighty-seven articles have been initially identified, of which 20 studies were excluded for various reasons, leaving, therefore, a total of 67 studies for our analysis. Response evaluation after the concurrent phase was performed in 36 (54%) studies but in only 14 (21%) response data were provided, while in 34 (51%) studies patients underwent a consolidation phase regardless of the response. No study provided any outcome (survivals, patterns of failure) as per response achieved after the concurrent phase. Conclusions: Information regarding the outcome of subgroups of patients achieving different responses after the concurrent phase and before the administration of the consolidation phase is still lacking. This may negatively affect the decision-making process as it remains unknown which patients may preferentially benefit from the consolidation of ST.

**Keywords:** locally advanced disease; inoperable nonsmall cell lung cancer; concurrent radiotherapy and systemic therapy; consolidation systemic therapy; chemotherapy; targeted therapy; immunotherapy





## 1. Introduction

Locally advanced inoperable nonsmall cell lung cancer (NSCLC) remains one of the major therapeutic challenges in thoracic oncology. While concurrent radiation therapy (RT) and systemic therapy (ST) (RT-ST) have been considered by many a standard treatment approach [1–3], results are still far from satisfactory, irrespective of the type of ST (chemotherapy, targeted therapy, immunotherapy). With the median survival times of 20–30 months and 5-year survivals of 25–30%, there is significant room for improvement. Various approach such as induction chemotherapy (CHT) followed by RT-CHT [4,5] or the addition of CHT or targeted therapy or immunotherapy after concurrent RT-ST [6–72] has been attempted. The latter one was intermittently labeled as either "maintenance" or "consolidation", with the majority of authors using the latter one.

Consolidation ST was seemingly first attempted some 26 years ago and has witnessed many transformations since then. During the initial years of its use, many studies used the same drugs given in both concurrent and consolidation phases of the treatment [6–13,15–17,19,21,22,25,28,29, 32–36,38,39,42,44,45,49,53,58,59,62,64,66–68,70] while many subsequent studies used different drugs ("switch" approach) [14,18,20,23,24,26,30,41,43,47,49,52,60,61] based on the assumption

that consolidation drugs may be more effective and less toxic when different from the concurrent ones. The recent decade also brought the slow but definitive introduction of both targeted agents [27,37,41,45,46,50,53,55–57,65], vaccines [40,51] and immunotherapy [63,69,71] in both the concurrent and consolidation phases of the treatment.

Regardless of the type and the number of drugs as well as the duration of the consolidation phase of the treatment, the evaluation of the patient response after the concurrent RT-ST phase of the treatment has been considered an important moment in the decision-making process, in which patients should continue the treatment with the consolidation phase [73–75]. While some studies did not undertake it [6,21,29,32,42,49,52,53,55,61,72], some did but continued with consolidation ST in all patients [7,8,11,15–20,30,31,37,38,40,44,45,48,51,56,59,60,62,66–68], while others mandated that the consolidation phase proceeds only in non-progressive disease (non-PD) patients [10,12–14,22–24,26,35,36,39,41,43,46,47,50,54,58,62,64,69–71]. In the latter case, non-PD included either a complete response (CR) or partial response (PR) but also stable disease (SD), with these three response groups having significantly different tumor burdens detected after concurrent RT-ST. The difference in the outcomes of these three patient groups may also have been observed if their outcomes were documented. Finally, possible differences between these groups may have indicated whether one or more of these may be unsuitable for consolidation treatment in case of significantly worse outcomes. Whether these issues were important to investigate, and possibly prove their influence on treatment outcomes remained largely unknown.

Based on these premises, we undertook the present analysis in order to investigate (1) treatment outcome per response (CR, PR, SD, PD) observed after receiving the concurrent phase of the treatment, specifically overall survival (OS), local progression-free survival (LPFS) and distant metastasis-free survival (DMFS), (2) exact (detailed) patterns of failure of patients receiving concurrent RT-ST, as per response achieved, in particular concerning the three patients groups (CR, PR, SD), and (3) treatment outcome per the pattern of failure occurring in one of these response categories. Taken together, the results achieved would help gather knowledge about which of these patients may or may not be suitable for such consolidation treatment in the future.

## 2. Materials and Methods

The information for this study was based on a search of the literature up to 1 August 2022 performed independently by three authors (BJ, EM, IS). It focused on the high-quality evidence, according to principles of evidence-based medicine, i.e., those of fully published prospective phase II and III studies which should have provided response rates and survivals as well as patterns of failure in the setting of concurrent RT-ST followed by consolidation ST. The search included published articles in the PubMed, SCOPUS, and Web of Science databases, and, once identified, their respective reference lists were checked for additional information/articles. The process was repeated with every newly identified and eligible study/reference. Meeting abstracts of potentially eligible studies were checked for a subsequent and full publication only if referenced in fully published articles and no systemic search was performed with those of various national or international meetings. The clinical trials registry database, ClinicalTrials.gov, was also searched for reports of closed/terminated, even if still unpublished trials. We variably combined the following keywords in our search: "non-small-cell lung cancer", "nonsmall-cell lung carcinoma", "nonsmall-cell lung neoplasm", "nonsmall-cell lung tumor", "nonsmall-cell lung tumour", "NSCLC", "radiochemotherapy", "chemoradiotherapy", "chemoradiation", "CCRT", "CRT", "targeted therapy", "immunotherapy", "stage III", "locally advanced", "inoperable", "maintenance", "consolidation", "phase II" and "phase III". We limited our search to full-length articles in English language journals. Published articles of phase II and phase III trials were selected for inclusion if they investigated the concurrent RT-ST in locally advanced (Stage III) NSCLC followed by consolidation ST. If a randomized study (phase II or phase III) contained one or more arms with other treatment options (e.g., induction ST followed by concurrent RT-ST), only the data from arm(s) using concurrent

RT-ST followed by a consolidation ST were used. Phase I-II studies were eligible if the phase II part was specified as per treatment details and sequence of the modalities as well as treatment outcome. Studies labeled as "pilot" were included if they provided the design we were seeking. No restriction was placed on either RT (total dose, dose per fraction, type of fractionation) or ST (type of drugs, number of drugs given either concurrently or in consolidation or the duration of consolidation treatment) characteristics. Studies were found ineligible if they were focused on recurrent lung cancer patients, those whose treatments included surgery or those that appeared only in the meeting abstract form. If a study produced more than one published article (e.g., updated with long term data) all articles from the same study were checked for the data, but only one article was included in the final list.

## 3. Results

Of the eighty-seven articles which have been initially identified as representing potentially eligible studies, six were excluded since they represented updates of previously published studies while two articles of a phase II study were also excluded due to the consolidation phase of the treatment not being an official part of the study or due to not all patients receiving it. Of the remaining 79 studies, potentially eligible for this report, two studies [76,77] were identified as meeting abstracts but we could not identify a subsequent full-length publication. Our attempt to communicate with authors, either personally or institutionally, gave no result. One study [78] had been fully published, but we had access to the abstract only as our attempt to obtain the full length article from the journal website or by directly communicating with authors was unsuccessful. Seven additional studies were identified existing on Clinicaltrials.gov having, however, no data pertinent to our study. They have been either closed or prematurely terminated but with no subsequent publication [79–85]; one of them being presented as a conference abstract [79]. Two studies [86,87], although both of a phase II design, were found ineligible due to a lack of necessary information from the consolidation part of the treatment. Therefore, a total of 67 studies have been identified as eligible for this report (Figure 1).

Eighty percent of all eligible studies were those including phase II while seventy-three percent of all studies were multi-institutional. While all studies included Stage III NSCLC patients, an occasional study also included either Stage II or Stage IV NSCLC patients. Sixty (89%) studies employed conventional RT with doses ranging from 40 to 74 Gy, while rare regimens included a hyperfractionated or split-course RT and one study did not provide RT data at all. Two studies were unique in study design: the SWOG study of Albain et al. [12] had the consolidation CHT part given with concurrent RT, making it thus, a split-course RT regimen, while in the PACIFIC study of Antonia et al. [63], more than 25% of patients received some form of CHT before the definitive concurrent RT-CHT part of the study. Only one study [40] used telomerase peptide vaccination in the consolidation arm, being grouped with other immunotherapy approaches (Table 1).

**Table 1.** Characteristics of eligible studies and treatments.

| Item | | N | % |
|---|---|---|---|
| Study type | Pilot study | 2 | 3 |
| | Phase I-II | 4 | 6 |
| | Phase II (single arm) | 42 | 63 |
| | Phase II (randomized) | 5 | 7 |
| | Phase III | 9 | 13 |
| | One arm from phase II | 3 | 5 |
| | One arm from phase III | 2 | 3 |
| Institution type | Single institution | 18 | 27 |
| | Multi-institutional | 49 | 73 |
| Stage of the disease | IIA-IIIB | 1 | 1 |
| | IIB-IIIB | 1 | 1 |

**Table 1.** *Cont.*

| Item | | N | % |
|---|---|---|---|
| | IIB-IIIC | 1 | 1 |
| | IIIA-IIIB | 57 | 85 |
| | IIIB | 4 | 6 |
| | III-IV | 3 | 5 |
| Type of RT fractionation * | Conventional (1.8–2.0 Gy/fx) | 60 | 89 |
| | Hfx (1.2–1.5 Gy/fx) | 4 | 6 |
| | Split course (1.8–3.0 Gy/fx) | 2 | 3 |
| | Not specified | 1 | 2 |
| RT total dose range per fractionation type * | Conventional: 40–74 Gy | 61 | 91 |
| | Hfx: 60–69.6 Gy | 4 | 6 |
| | Split course: 60–61 Gy | 2 | 3 |
| | Not specified: not specified | 1 | 1 |
| No. drugs given concurrently with RT * | 1 drug | 8 | 11 |
| | 2 drugs | 53 | 74 |
| | ≥3 drugs | 11 | 15 |
| Type of drugs given concurrently with RT * | CHT alone | 58 | 82 |
| | Targeted alone or in combination | 7 | 10 |
| | Immunotherapy alone or in combination | 2 | 3 |
| | CHT with non-anti-cancer drugs | 4 | 6 |
| No. drugs given in consolidation phase * | 1 drug | 21 | 30 |
| | 2 drugs | 40 | 56 |
| | ≥3 drugs | 11 | 15 |
| Type of drugs given in consolidation phase * | CHT alone | 54 | 78 |
| | Targeted alone or in combination | 6 | 9 |
| | Immunotherapy alone or in combination | 6 | 9 |
| | CHT with non-anti-cancer drugs | 3 | 4 |
| Drugs given in consolidation phase * | Same as in concurrent phase | 31 | 46 |
| | Different (switch) | 20 | 29 |
| | >1 drug remains the same | 17 | 25 |
| Duration of the consolidation phase | 2 cycles | 31 | 46 |
| | 3 cycles | 18 | 27 |
| | 2–4 | 1 | 2 |
| | 3–5 cycles | 1 | 2 |
| | 4 cycles | 8 | 12 |
| | Prolonged administration | 8 | 12 |

* = more than one option existing; Gy = Grey; fx = fraction; Hfx = hyperfraction; CHT = chemotherapy.

Seventy-four percent of all eligible studies used two-drug regimens for the concurrent phase and in 82% of studies patients received CHT alone. In the consolidation phase, in 56% of studies patients received two drugs, and in 78% of studies, patients received CHT alone. In less than 50% of studies, patients received the same drugs in a consolidation phase, but at least one drug in the consolidation phase of the treatment remained the same as in the concurrent phase in 25% of studies. Less than 50% of all eligible studies witnessed the consolidation phase of the treatment using two cycles, slightly more than one-fourth received three cycles, while prolonged administration of the newest compounds (targeted agents and immunotherapy) was observed in the most recent studies and was seen in 12% of studies.

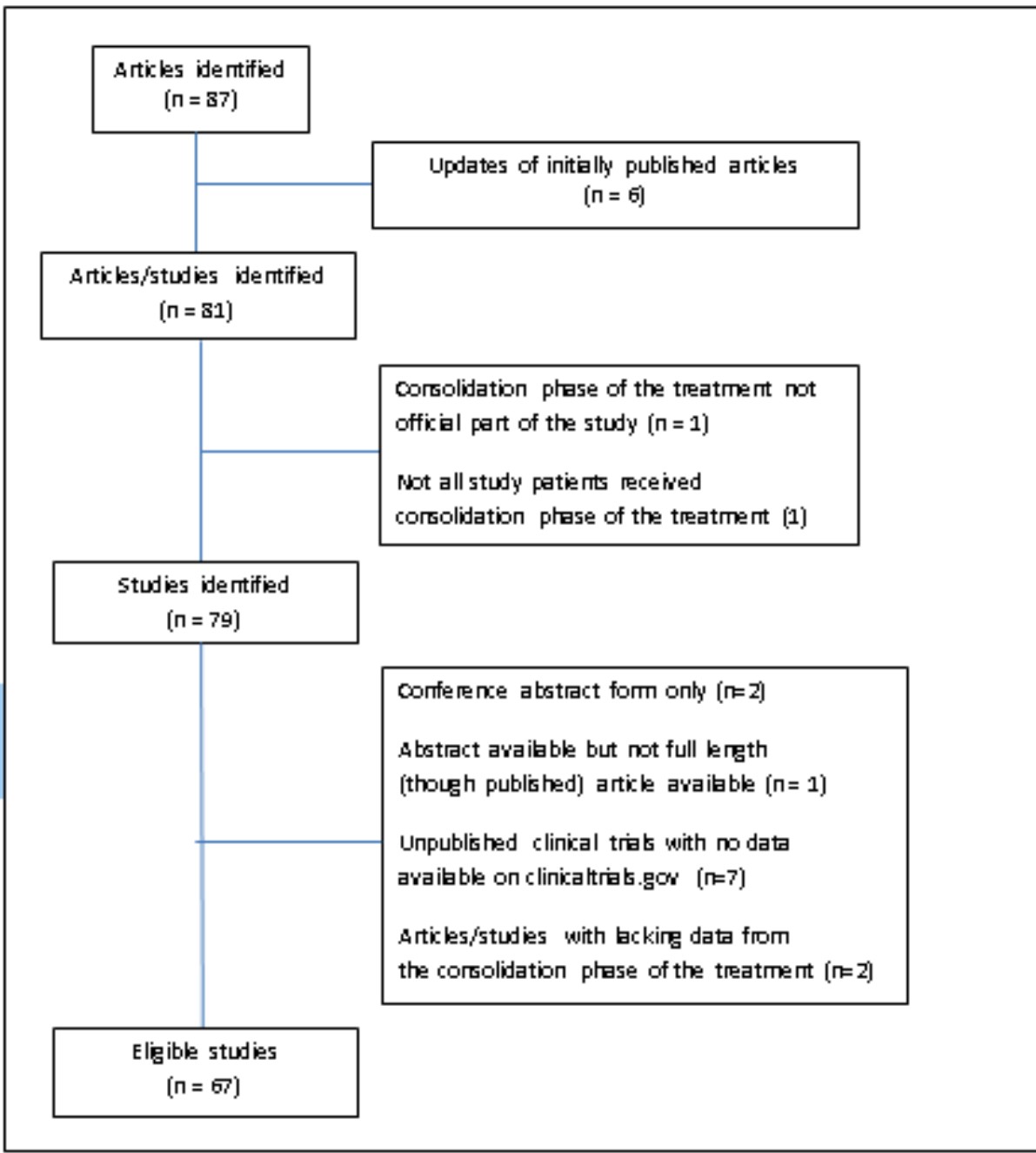

**Figure 1.** Flow chart of the search process with 67 eligible and 20 ineligible records.

When the time gap between the end of the concurrent RT-ST phase and the start of the consolidation ST phase of the treatment was evaluated, no gap was observed in 12% of the studies, and with the same percent of studies not reporting it. Thirty-six percent of all studies used rather "flexible" time gaps after the concurrent phase of the treatment. Although the response evaluation after the concurrent phase was conducted in more than 50% of studies, their results have been provided in only 21% of all studies. When the type of patient (CR, PR, SD or PD) as a potential candidate for continuing the treatment with the consolidation phase was further investigated, we found that approximately one-half of the studies have the consolidation phase administered irrespective of the response status of the patients while in three studies this matter was not specified at all. Response after the treatment, reported either after the whole treatment or as "the best response" was reported in almost 50% of the studies (Table 2).

**Table 2.** Characteristics and results of evaluation.

| Characteristic | | N | % |
|---|---|---|---|
| Time gap between concurrent and consolidation phase | No gap | 8 | 12 |
| | 2 weeks | 3 | 4 |
| | 3 weeks | 9 | 13 |
| | 4 weeks | 15 | 23 |
| | Various | 24 | 36 |
| | unknown | 8 | 12 |
| Evaluation conducted after concurrent part | Yes | 36 | 54 |
| | No | 31 | 46 |
| Response after concurrent part provided | Yes | 14 | 21 |
| | No | 53 | 79 |
| Response provided after the whole course of the treatment | Yes | 32 | 48 |
| | No | 35 | 52 |
| Type of patients continuing with consolidation treatment | All | 34 | 51 |
| | Non-PD | 28 | 42 |
| | Responders | 2 | 3 |
| | Not specified | 3 | 4 |

PD = progressive disease.

No study provided an outcome as per the response achieved after the concurrent RT-ST phase of the treatment. Only the PACIFIC study [63] provided a comparison of the outcome for PR versus SD patients after the concurrent RT-ST phase of the treatment, but only as unstratified hazard ratios for disease progression or death and without any further details. While some studies reported "basic patterns of failure" (local, and distant) and some others provided more details (e.g., per organ/site of failure) for the whole patient population, no study provided patterns of failure for various response categories achieved after concurrent RT-ST phase of the treatment. As a consequence, no study provided the outcome after the failure occurred in any of the response categories.

## 4. Discussion

The history of consolidation ST given after concurrent RT-ST witnessed many changes. Initial studies used the same drugs in both phases of the treatment, presumably building upon the issue of adding more CHT in the consolidation part. This approach neglected the possibility that the same consolidation drugs brought nothing but more toxicity due to more drug doses being given. Importantly, the lack of effectiveness of such an approach in patients whose tumors may already have developed resistance to drugs given concurrently with RT was not taken properly into account.

However, once this was conceptualized, subsequent studies used the so-called "switch" approach, i.e., drugs given in the consolidation phase differed from those given in the concurrent phase. That way, one would avoid drug resistance and enable the consolidation drugs to act both intrathoracically as well as at presumably existing micrometastases, hopefully not increasing toxicity. Further attempts included targeted agents and then immunotherapy. Although some phase II studies brought promising results, those of phase III design [26,27] as well as pooled analysis [88] and the meta-analysis [89] showed no advantage of the consolidation approach over the exclusive concurrent RT-ST.

However, one meta-analysis [90] showed a benefit for the consolidation ST over the concurrent RT-ST in terms of OS, but not on PFS or ORR. Ten years ago, another meta-analysis [91] investigated maintenance therapy with either a continuous or a switch strategy for patients with non-progressing NSCLC compared with the placebo or observation and showed that switch maintenance therapy substantially improved OS compared with placebo or observation. A similar trend of improved OS was found in continuous maintenance therapy, despite lacking statistical significance. The interaction test suggested that the difference in OS between the two maintenance strategies was not statistically significant. Clinically substantial and statistically significant improvement in PFS was

found with both maintenance strategies. Subgroup analyses revealed no statistically significant differences in OS or PFS between switch maintenance therapy with cytotoxic agents and that with tyrosine kinase inhibitors. Toxicity was greater in maintenance therapy. Despite these uncertainties, many researchers continued practicing consolidation therapy and it was only a recent PACIFIC study that showed for the first time, the effectiveness of consolidation immunotherapy after concurrent RT-CHT [63].

Whichever approach one considers over time, authors were frequently discussing the "optimization" of the approach but unequivocally remained focused on the choice of the drugs, the intensity of ST and cross or non-cross drug resistance. No discussion and consideration were undertaken to enlighten the problem of choice of appropriate patients deemed suitable for consolidation ST, except perhaps differentiating between non-PD and PD patients. The question, therefore, remained unanswered as to who are the patients which may benefit from consolidation ST and if there are several potential groups of interest, how one can discriminate between them, eventually and hopefully, quantifying the benefit of such a treatment approach. In our study we expected to gain information about who should preferentially continue with the consolidation ST and, if properly investigated, how and where consolidation ST acts. The last question had an important aspect when one knows that in some studies all patients continued consolidation ST while in some it was non-PD patients. However, if we assume that fewer studies are using the former approach nowadays, we are still left with three possible types of patients seen after the concurrent phase, namely those with either CR or PR or SD. Discrimination between the outcomes of these three subgroups of non-PD patients may be important from several standpoints.

First and foremost, these three patient subgroups have different disease burdens at the time of response evaluation conducted after the concurrent phase: those with CR have only microscopic disease intrathoracically and possibly elsewhere, while those with either PR or SD have present (though potentially of a large difference) intrathoracic disease and possible microscopic disease. Regardless of the action of drugs given in a consolidation phase, it is reasonable to assume that it would be easier for an effective ST to eradicate microscopic deposits existing intrathoracically or elsewhere. Whether all or some patients with either PR or SD should receive consolidation ST would also depend on a number of factors such as the choice of drugs and their administration (doses, duration, etc.).

Second, by observing the patterns of failure in each of the three subcategories of non-PD patients, one would be able to learn where those patients actually fail with a specific ST. In other words, and only as a hypothesis, perhaps patients achieving CR after the concurrent phase and any consolidation ST would be failing more often at distant sites than intrathoracically, indicating, perhaps that "switch" therapy may be preferential to continuation with the same drugs.

Third, and to prove the above using LPFS and the DMFS as per the achieved response after the concurrent phase in these three subgroups could give us information about the tempo of such developments and, perhaps, indicate the duration of the consolidation ST in one or more of these subgroups.

Unfortunately, the literature does not offer such insight. No study ever investigated treatment outcome per response (CR, PR, SD, and PD) observed after receiving the consolidation part of the treatment, specifically OS, LPFS and DMFS. The only study which provided a hint towards this issue was the PACIFIC study [63] which showed benefits for both patients with PR or SD for the immunotherapy agent Durvalumab, and a similar outcome for the two response categories. Additionally, no study ever provided exact patterns of failure of patients receiving concurrent RT-ST, in particular, concerning the three patient groups (CR, PR, SD), and consequently, the treatment outcome per pattern of failure occurring in one of these response categories remained unknown. We are, therefore, left without the results which could have helped us gather the knowledge about which of these patients may or may not be suitable for such consolidation ST in the future.

This unfortunate finding, however, enables us to speculate about possible scenarios in patients experiencing different responses after the concurrent phase of the treatment.

For example, in patients achieving a CR after the concurrent phase, though they are observed much less frequently than either PRs or SDs, do they need a consolidation ST at all? If yes, whether one should continue with the same drugs as those given in the concurrent phase since achieved CR could be seen as the "proof" of drug activity. Perhaps switching to different drugs in the consolidation part presumably due to locoregional, later-appearing, recurrences presumably originated from drug-resistant clones. In patients with "good" PR, though exactly how "good" is needed remains questionable, perhaps there may be an interesting research pathway for adding a stereotactic body RT (SBRT), due to RTOG0617 [55] effectively re-tuning the RT dose to the 60 Gy level, leaving some space (i.e., additional RT dose) to be given. This may especially be interesting in cases where the N component and achieved CR and T component achieved significant size reduction (i.e., PR). Additionally, in both patients with PR or SD, perhaps a salvage surgery could be attempted. Finally, in the latter group, those with SD, should one expect that after 60 Gy and 2–3 cycles of concurrent ST consolidation treatment would substantially change the fate of such patients, at least for the present times as the PACIFIC study results [63] hinted.

Although our efforts were unsuccessful, several items could have played a role in better understanding the whole issue, but nevertheless, they remain important aspects of future research in this field. One of these is the fact that response evaluation criteria varied from study to study. While some described it without any reference to existing systems [7,8,10], others used those of the Eastern Cooperative Oncology Group [9], World Health Organization [11,12,16,18], or cancer and Leukemia Group B [17]. It must be, however, stressed that the most recent attempt used Response and Evaluation Criteria In Solid Tumors (RECIST) more consistently, with the hope that uniformed response criteria should bring more uniformity in response reporting. The time of patient enrollment in the respective studies may also be critical and warrant further investigation. The most valid studies to include may well be those that enroll patients before concurrent radiotherapy and systemic therapy. Studies that enroll patients after radiotherapy, on the other side, may reflect a different population, food for thought for researchers in this field. Additionally, clinical observations of the natural course of intrathoracic tumors point to the fact that once a lesion is irradiated, it can continue to decrease in size even in the absence of further systemic therapy. This is why a lesion that has been radiated may not be measurable for response until it shows growth. Therefore, the timing of the response evaluation after concurrent RT and systemic therapy may be an important aspect to consider when planning a study or evaluating its results.

## 5. Conclusions

Our study, the first ever with this aim, reviewed the highest level of evidence, those of phase II and III studies in the setting of concurrent RT-ST followed by a consolidation ST in patients with locally advanced inoperable NSCLC. It covered a period of 26 years of available and fully published data in the English language literature with a total of 67 studies being eligible for this investigation. Our study brought evidence of systematic neglect of an important issue in this setting: there is not a single study that investigated either the treatment outcome per response observed after receiving the concurrent phase of the treatment, or exact patterns of failure of patients receiving concurrent RT-ST as per response achieved, or treatment outcome per the pattern of failure occurring in one of these response categories. Thoracic oncologists are, therefore, urged to attempt to provide answers to some if not all of these questions, evaluating their own data both retrospectively and prospectively as well as taking these issues into account when planning future studies with consolidation ST given after concurrent RT-ST.

**Author Contributions:** Conceptualization, B.J. and I.K.; methodology, B.J., E.M. and I.S.; validation, B.J., E.M. and I.S.; formal analysis, B.J., E.M. and I.S.; investigation, B.J., E.M. and I.S.; writing—original draft preparation, B.J. and I.K.; writing—review and editing, B.J., E.M., I.S. and I.K.; supervision, B.J. and I.K.; project administration, B.J. and I.K. All authors have read and agreed to the published version of the manuscript.

**Funding:** This research received no external funding.

**Institutional Review Board Statement:** This study did not require ethical approval.

**Informed Consent Statement:** Not applicable.

**Conflicts of Interest:** The authors declare no conflict of interest.

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
