# Peer review of "Consolidation Systemic Therapy in Locally Advanced, Inoperable Nonsmall Cell Lung Cancer—How to Identify Patients Which Can Benefit from It?"

_curroncol, doi:10.3390/curroncol29110656_

Round 1

Reviewer 1 Report

Dr. Jeremic et al present a systematic review of the literature exploring how response to concurrent systemic therapy + radiotherapy influences outcomes with subsequent consolidation therapy. Their review identified 87 trials, ultimately including 67 in their analysis. The conclusion is that the data are simply not available from these studies. This is the contribution of the work and if published, the manuscript can serve to raise awareness of the need for these data and prevent others from seeking to answer the same question. I think this is a reasonable contribution to the field that warrants publication, but there are some points that need to be made.

1.     The title should be more descriptive of the work done so that others seeking to ask the same question will find this manuscript. The current title is more appropriate for an editorial and not a systematic review.

2.     The references need to be numbered in the order they appear in the text.

3.     The authors are categorizing outcomes based on response to initial systemic therapy + radiotherapy. As this is the key variable, it is important to properly define “response.” Were all of the responses in the studies using “RECSIT” criteria or just investigator-report? That needs to be clarified as they are two very different variables.

4.     When referencing the “flexible” gap between radiotherapy and consolidation, the authors note that some had a gap of 24-48 weeks. I encourage the authors to verify the 48 weeks, as this is almost a year later and seems incorrect.

5.     The authors note the response to consolidation therapy after concurrent radiotherapy with systemic therapy and include it in the table but this needs a closer look. Once a lesion is radiated, it can continue to decrease in size even in the absence of further systemic therapy. This is why a lesion that has been radiated is not measurable for response until it shows growth. This is a criticism of PACIFIC. I suggest removing reference to this variable.

6.     Finally, the time of patient enrollment on the respective studies is critical and warrants mention. The most valid studies to include are those that enroll patients before concurrent radiotherapy + systemic therapy. Studies that enroll patients after radiotherapy will reflect a different population. For example, PACIFIC only enrolled patients after chemoradiation and to be eligible, patients could not have had progression after CRT. This would skew the analysis, if it would have been able to be completed.

Author Response

Item #1 - we changed the article title as suggested - new title reads as 

"CONSOLIDATION SYSTEMIC THERAPY IN LOCALLY ADVANCED, INOPERABLE NONSMALL CELL LUNG CANCER. HOW TO IDENTIFY PATIENTS WHICH CAN BENEFIT FROM IT?"

Item #2 - we double-checked all references as they appear in the text and could not find any case when we have not followed journal isntruction (to have therm as numbered they first appear in the text)

Items #3, 5, and 6 - sicne we agree with the reviewer about importance of these issues, we have introduced them in a special paragraph at the end of the discussion section

Item #4 - we are grateful to this reviewer for spotting this item and suggestng we double check it. After checking it, we have excluded the words related to "various gap times" from the text

Reviewer 2 Report

The manuscript is a literature analysis aiming to find out which patients with lung cancer would benefit from the additional therapy, and which would not. The manuscript is well-written, and well-structured, as well as raises important questions in the research field that need answers during the recent decades.

1) Figure 1 has no Figure legend and the Figure itself is too extended. It could look as good using half-page of the space, and the Figure legend could summarize and extend the information presented in the Figure.

2) The first paragraph of the discussion is rather long. It might be beneficial for the readers to have smaller paragraphs. 

3) Importantly, only about 10% or references are recent. In addition to the work done 10-15-20 years ago, it would be beneficial to add several fresh references, including 2022.

Author Response

Item #1 - we resheped the Figure 1 and added a Legend as requested

Item #2 - we reshaped the first paragraph of the Discussion section by splitting into three smaller ones, as suggested

Item #3 - while we agree with this reviewer that the vast minority of the references are recent ones, they are as they are: our search started with those originating in 1996 and followed articles appearences till nowadays with observation that very few studies were of recent origin. If this reviewer, however, knows some reference(s) which we missed, we would be more than happy to include it/them in re-revision